# Numerical Simulation of Nitrogen-Doped Titanium Dioxide as an Inorganic Hole Transport Layer in Mixed Halide Perovskite Structures Using SCAPS 1-D

**Nitin Ralph Pochont** [1] and **Yendaluru Raja Sekhar** [2,*]

1   School of Mechanical Engineering, Vellore Institute of Technology, Vellore 632014, India
2   Centre for Disaster Mitigation and Management, Vellore Institute of Technology, Vellore 632014, India
*   Correspondence: rajasekhar.y@vit.ac.in

**Abstract:** Perovskite solar cells (PSCs) stand out as superior third-generation (III-gen) thin-film energy harvesting structures with high efficiency, optical properties and light transmission ability. However, the need to develop cost-effective, stable and sustainable PSCs is allied to the influence of the absorber layer and charge selective transport layers when achieving semi-transparent (ST) structures. Using SCAPS simulation software that can envisage the conceptuality in devising ST PSCs, this work explores and reports the electrical performance of different methylammonium (MA)-based perovskite structures (FTO/TiO$_2$/PCBM/SnO$_2$/MAPbI$_3$/TiO$_2$:N/PTAA/Spiro-OMeTAD/PEDOT: PSS/Ag). The influence of absorber thickness and defect density is analyzed with optimal parameters. This research reports a novel idea that replaces the polymeric hole transport layer (HTL), such as Spiro-OMeTAD, PEDOT: PSS and PTAA with an air-stable inorganic metal oxide, viz., nitrogen-doped titanium dioxide (TiO$_2$:N). The simulation results depict an attainable power conversion efficiency of 9.92%, 10.11% and 11.54% for the proposed structures with the novel HTL that are on par with polymeric HTLs. Furthermore, the maximum allowable absorber thickness was 600 nm with a threshold defect density of $1 \times 10^{15}$ cm$^{-3}$. The optimized electrical parameters can be implemented to develop thin-film light transmission perovskite cells with rational power conversion efficiencies.

**Keywords:** nitrogen-doped titanium dioxide; inorganic layer; absorber thickness; defect density; electrical parameters; SCAPS simulation; semi-transparent perovskite cells





## 1. Introduction

### 1.1. An Overview of Perovskite Solar Cells

Solar irradiance offers a clean mode of energy production with broad engineering applications. To aspire to the energy sustenance goals for 2030, net-zero emission concepts through renewable sources will foster the achievement of the set targets [1]. In this regard, solar photovoltaics is known to be a prominent renewable source with potential commercial markets. The evolution of solar cells using sophisticated material physics has led to a technological revolution for developing energy generation solutions in diverse applications such as buildings, aircraft and satellites. Photovoltaic solar cells are classified into distinct generations, based on semiconductor raw materials and fabrication techniques, as well as their era of evolution [2]. Substantially, the advent of third-generation solar cells (III-gen SCs) has changed the phase of photovoltaics due to higher power conversion efficiency reaching up to 30%. The significant advantage of III-gen SCs is that one can achieve transparency or semi-transparency, which has paved remarkable novel applications in source integrations [3]. Amongst the different III-gen SCs, perovskite solar cells (PSCs) have gained tremendous interest in the last decade due to their high efficiencies compared to their other counterparts. PSCs' optical and electrical properties include direct and tunable bandgap, high absorption coefficient and longer charge carrier diffusion length [4]. Researchers have developed unique recipes that achieved a power conversion efficiency

(PCE) of 25.2%, which was the apex reported in 2020 [5]. Further findings reveal that PSCs can reach an efficiency of 26.1%, and a multi-junction PSC with a Si-tandem structure attained a PCE of 29.8% [2].

Over the years, the perovskite $ABX_3$ structure has evolved into a concoction of various elemental combinations [6]. The multi-element $ABX_3$ system of a PSC comprises diverse metal halide organic and inorganic elements, Figure 1. Methylammonium lead halide ($MAPbX_3$) and formamidinium lead halide ($FAPbX_3$) are the most endorsed organic elements in formulating the cell recipe in planar or mesoporous structures. On the other hand, cationic organic elements (MA and FA) have depicted an intrinsically unstable behavior which led to the presage of inorganic cationic elements such as cesium (Cs), rubidium (Rb), francium (Fr), potassium (K), respectively [7,8]. On the other hand, metals (B) and anions (X) are engineered with different organic and inorganic elements because of enhancing stability, light transmission, and cell efficiency [9,10]. Moreover, within the metal elements, the presence of lead (Pb) increases the toxicity of the cell. Henceforth, literature reports several group IVA and VA elements such as tin (Sn), germanium (Ge), antimony (Sb), bismuth (Bi) and silicon (Si) as an ideal replacement in several chemical PSC recipes [4,11,12]. Lastly, the range of anionic elements revolves majorly around chlorine ($Cl^-$), bromine ($Br^-$) and iodine ($I^-$) [13]. While the top conducting electrodes are generally gold (Au) or silver (Ag), and the bottom substrate is an indium-doped tin oxide (ITO) or a fluorine-doped tin oxide (FTO) glass.

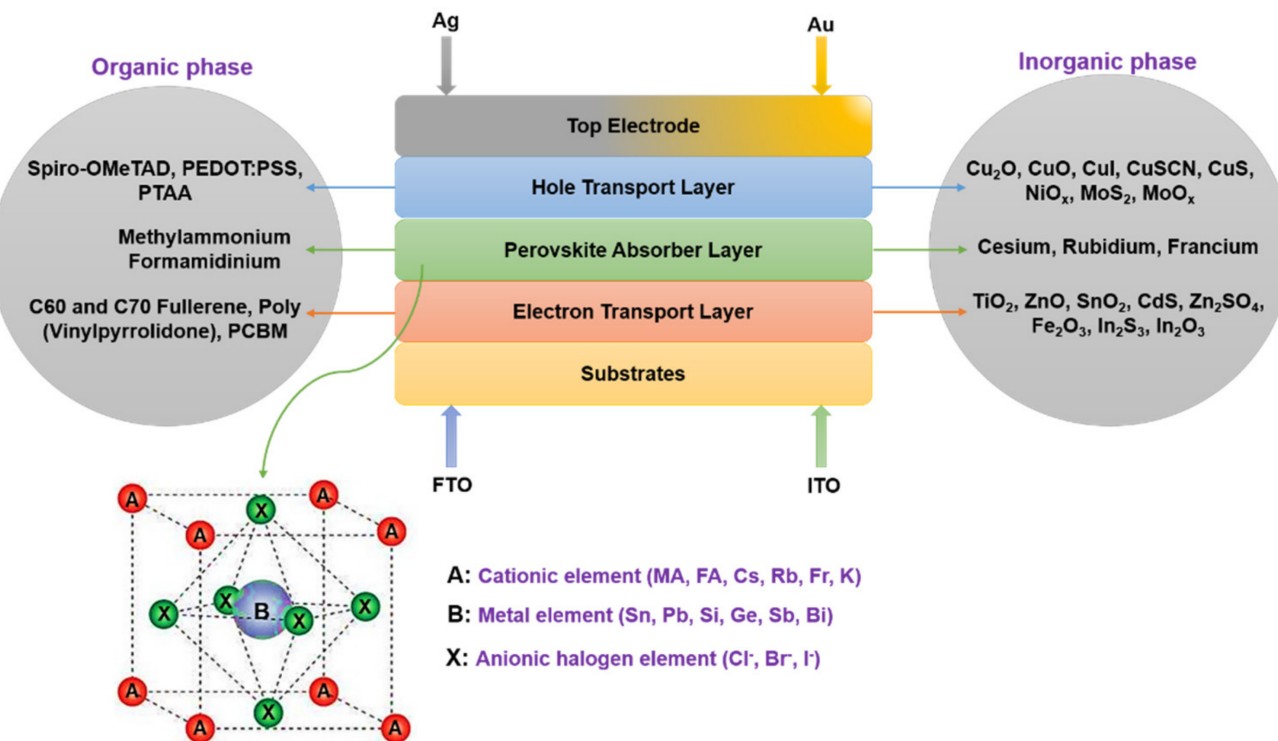

**Figure 1.** The multi-element structure of a perovskite solar cell.

Charge transport layers in the PSC structure deliver the carrier medium for holes and electrons. The absorber layer is sandwiched between the electron transport layer (ETL) and the hole transport layer (HTL) [14]. The physics behind the transport layers depicts their role in avoiding recombination, preventing cell degradation, boosting charge transportation and efficient light transmission characteristics. The intrinsic absorber layer and the charge transport p-type and n-type layers club to formulate a p-i-n or an n-i-p structure [15]. The transport layers are also deemed to act as blocking layers to their opposite counterparts. The HTLs and ETLs are chosen based on the organic and inorganic material band structure and band edge position. Organic HTLs are advantageous in biodegradability and layer

processing [16], while the drawbacks incurred due to instability, high cost and multi-step synthesis have led to the adoption of inorganic HTLs. In addition, inorganic HTLs promote appealing features such as hole mobility, chemical stability and low-cost [17].

On the other hand, ETLs are chosen based on efficient electron extraction ability and stability. The most common ETLs preferred in a PSC structure are the inorganic-based $TiO_2$, ZnO and $SnO_2$ [18] because of their surface and electrical properties. Furthermore, organic ETLs have less emphasis on the performance output when compared to inorganic ETLs.

### 1.2. Semi-Transparent Perovskite Solar Cells

Perovskite solar cells acquire exclusive electrical and optical properties such as long charge carrier diffusion length and high absorption coefficient with a direct and tunable bandgap [3]. The organic–inorganic layers in a PSC structure provide an interfacial synergy that nurtures the ability to develop semi-transparent (ST) solar cells. ST PSCs are wavelength perspective structures with the interest of spectral absorbance focused on specific wavelengths in the entire spectrum. In response to the human eye, the average light transparency in the visible range of 400 to 700 nm defines a dimensionless parameter called the average visible transmission (AVT) [19]. The AVT is calculated with the relation [19,20]

$$AVT = \frac{\int T(\lambda).P(\lambda).I(\lambda).d(\lambda)}{\int T(\lambda).I(\lambda).d(\lambda)}$$

where λ—wavelength, T—transmission spectrum, P—photopic response, I—solar flux.

The absorption coefficient in a PSC absorber layer is interlaced with its thickness, which is deemed to reduce to achieve better light transmission properties. However, the reduction in thickness reflects in the drop of open circuit voltage ($V_{oc}$) and short-circuit current ($J_{sc}$) [21]. Intuitively, the power conversion efficiency (PCE) holds an inversely proportional behavior to the AVT due to the constraint caused to thinner interfacial layers and wavelength perspective absorbance. Research reports that the PCE is observed to reduce with the increase in the AVT and vice-versa. The key takeaways and strategies that offer the possibilities to attain ST PSCs are illustrated in Figure 2.

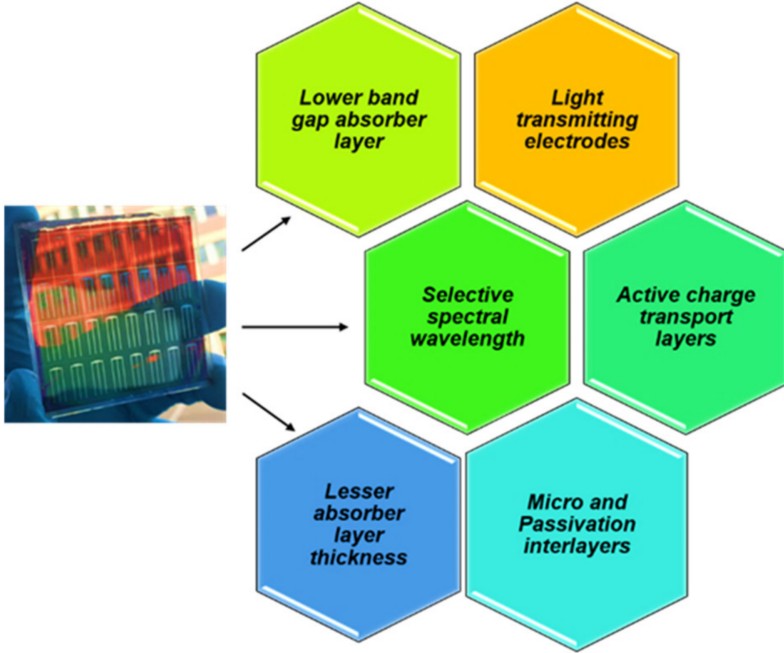

**Figure 2.** Key takeaways and strategies to attain ST perovskite solar cells [22].

### 1.3. The Absorber and Charge Transport Layers in an ST PSC

The feasibility of fabricating a stable and efficient ST PSC is possible with molecular engineering of the absorber layer, charge transport layer and conducting electrodes. To achieve efficient electrical and optical properties, researchers worldwide have devised various recipes with a diverse use of elements and their molecular combinations. The absorber layer plays a prime role in delivering efficient outputs. Amongst the different organic/inorganic absorber layers, organic methylammonium lead iodide ($MAPbX_3$) is considered one of the viable layers that suit the structure of an ST PSC. The modulation of band gap and thickness offers broad characteristics to envisage high electrical properties and light transmission [23]. At even the lowest absorber thickness of 50 nm, $MAPbX_3$ provides a stable and homogenous layer that can foster high efficiency and high visible light transmission properties [24]. Secondly, the charge transport layers are responsible for the subsistence flow of holes and electrons with no scope for recombination. Due to high light transmission properties, PEDOT: PSS, PTAA and Spiro-OMeTAD are well-known HTLs employed in achieving ST PSCs [25,26]. While visible light-active carbon-loaded anatase titanium dioxide ($c-TiO_2$) and organic fullerene derivative [6,6]-phenyl-C61-butyric acid methyl ester (PCBM) are well known to have been employed as ETLs to fabricate an ST PSC. The bottom substrates (ITO or FTO) are directly used in the cell structure, but research also reports the possibility of doping the glass substrates with zinc to enhance its strength and bidirectional properties [27–33]. Finally, top electrodes Au or Ag are commonly used, while Au offers better stability from oxidation and a low halide reactivity compared to Ag. Table 1 reports the different ST PSC structures developed in the past decade.

**Table 1.** ST PSC structures developed in the past decade.

| S. No. | Author | ST PSC Device Structure (Top Electrode/HTL/Absorber Layer/ETL/Substrate) | PCE (%) | AVT (%) | Reference | Year |
|---|---|---|---|---|---|---|
| 1 | Matteocci et al. | $ITO/PTAA/MAPbBr_{1-x}Cl_x/c-TiO_2/m-TiO_2/FTO$ | 64.9 | 6.3 | [28] | 2022 |
| 2 | Ponchai et al. | $Au/Spiro-OMeTAD/(PEA)_2MA_{n-1}Pb_nBr_{3n+1}/TiO_2/FTO$ | 4.86 | 26.12 | [29] | 2021 |
| 3 | Rani et al. | $Ag/NiO_x/MAPbI_3/TiO_2/ZnO/ITO$ | - | 31.49 | [30] | 2021 |
| 4 | Singh et al. | $Au/PTAA/MAPbBr_3/c-TiO_2/mp-TiO_2/FTO$ | 7.6 | 52 | [31] | 2021 |
| 5 | Zhang et al. | $Ag/Zr/PCBM/MAPbI_3/NiO_x/ITO$ | 11.74 | 23 | [32] | 2021 |
| 6 | Dew et al. | $ITO/Ag/PEDOT: PSS/MAPbI_3/PC_{61}BM/ITO$ | 7.4 | 8 | [33] | 2019 |
| 7 | Yuan et al. | $Ag/ZnO/PC_{61}M/MAPbI_3/PEDOT: PSS/ITO$ | 8.5 | 28.4 | [22] | 2018 |
| 8 | Han et al. | $Ag/PCBM/MAPbI_{3-x}Cl_x/PEDOT: PSS/ITO$ | 13.27 | 16.3 | [34] | 2018 |
| 9 | Chen et al. | $Au/Spiro-OMeTAD/MAPbI_3/TiO_2/FTO$ | 11.7 | 36 | [35] | 2016 |
| 10 | Hörantner et al. | $Au/Spiro-OMeTAD/MAPbI_3/c-TiO_2/FTO$ | 6.1 | 38 | [36] | 2016 |
| 11 | Lee et al. | $Ag/PEDOT: PSS/MAPbI_3/PCBM/C_{60}/ITO$ | 8.2 | 34 | [37] | 2016 |
| 12 | Chang et al. | $Ag/PEDOT: PSS/MAPbI_3/PC_{61}BM/ITO$ | 11.8 | 20.8 | [38] | 2016 |
| 13 | Heo et al. | $Ag/ZnO/PC_{61}BM/MAPbI_{3-x}Cl_x/PEDOT: PSS/ITO$ | 7.8 | 37 | [39] | 2016 |
| 14 | Heo et al. | $Au/Spiro-OMeTAD/MAPbI_3/c-TiO_2/mp-TiO_2/FTO$ | 4.9 | 19 | [40] | 2015 |
| 15 | Della Gaspera et al. | $SnO_x/Ag/PEDOT: PSS/MAPbI_3/PCBM/ITO$ | 11.8 | 29 | [24] | 2015 |
| 16 | Eperon et al. | $ITO/PEDOT: PSS/PTAA/MAPbI_3/TiO_2/FTO$ | 10.6 | 20.9 | [41] | 2014 |
| 17 | Lynn et al. | $Au/Spiro-OMeTAD/MAPbI_3/c-TiO_2/FTO$ | 3.5 | 30 | [42] | 2012 |

### 1.4. Nitrogen-Doped Titanium Dioxide—A Proposed Novel HTL

Titanium dioxide ($TiO_2$) emerges as a prevalent low-cost photocatalyst with chemical stability and nontoxicity [43]. The cationic or anionic doping modifies this material's bandgap, optical and electrical properties [44]. With tunable bandgap and fermi-level shift, $TiO_2$ doped with various elements has been demonstrated to be a promising charge transport layer with improved efficiency and open-circuit voltage ($V_{oc}$) [45]. Literature reports $TiO_2$ as an ETL in the PSC recipe, but this transport layer perseveres with a limitation of poor visible light absorption. Nitrogen is a suitable doping element in $TiO_2$ that forms a metastable center, reduced atom size, and low ionization energy [46]. In the recent past, Panepinto et al. [47] reported a breakthrough in devising a nitrogen-doped $TiO_2$ ($TiO_2$:N)

layer as an HTL for application in dye-sensitized solar cells. The $TiO_2$:N HTL layer was fabricated by co-reactive magnetron sputtering and tuning of $O_2$ and $N_2$ reactive gas mixture. The doping concentration (%) of nitrogen determines the tunable bandgap of the HTL. Although the usage of $TiO_2$:N as a low-cost HTL in perovskite solar cell structures has not been explored so far, with enhanced photocatalytic properties, the applicability of $TiO_2$:N as a stable and low-cost HTL is suitable for semi-transparent PSCs.

### 1.5. Overview of This Research

This research aims to simulate the charge characteristics and reports the electrical performance of a proposed ST PSC structure with $TiO_2$:N as the HTL medium. The literature reported in Table 1 quantifies the persistent ETLs ($TiO_2$, PCBM and $SnO_2$) and polymeric HTL (Spiro-OMeTAD, PEDOT: PSS and PTAA) mediums used in an ST PSC structure, while oxide-based HTL remains scarce. Since the polymeric HTLs are unstable in nature and high in cost, adopting $TiO_2$:N as an HTL would deliver an air-stable charge transport medium; this concept has not been performed earlier. Henceforth, the idea of simulating a planar $MAPbI_3$-based perovskite structure with a comparative analysis (nine different structures) is reported through this research. This study investigates the suitability of the proposed HTL and reports the results obtained by choosing the optimum electrical parameters that can be used when developing an ST PSC practically. These simulation results can be inherited to fabricate a stable and efficient PSC structure with light transmission properties.

### 2. Results and Discussion

Considering the electrical parameters proposed in the methodology from Section 3.3, with no resistances and dark current, these numerical simulations were performed to estimate the performance output for the proposed PSC. The results attained through SCAPS simulation are presented in Table 2, while the current–voltage characteristics of the PSC structures are displayed in Figure 3a–c.

**Table 2.** Electrical parametric output attained from the simulation results.

| Device Structure | $V_{oc}$ (V) | $J_{sc}$ (mA/cm$^2$) | FF (%) | PCE (%) |
|---|---|---|---|---|
| FTO/$TiO_2$/$MAPbI_3$/$TiO_2$:N/Ag | 1.07 | 11.35 | 81.09 | 9.92 |
| FTO/$TiO_2$/$MAPbI_3$/Spiro-OMeTAD/Ag | 1.10 | 11.30 | 80.30 | 10.00 |
| FTO/$TiO_2$/$MAPbI_3$/PTAA/Ag | 1.08 | 11.32 | 82.00 | 10.12 |
| FTO/$TiO_2$/$MAPbI_3$/PEDOT: PSS/Ag | 1.08 | 13.09 | 83.00 | 11.75 |
| FTO/PCBM/$MAPbI_3$/$TiO_2$:N/Ag | 1.10 | 12.48 | 83.32 | 11.54 |
| FTO/PCBM/$MAPbI_3$/PTAA/Ag | 1.12 | 12.46 | 83.75 | 11.73 |
| FTO/PCBM/$MAPbI_3$/Spiro-OMeTAD/Ag | 1.13 | 12.44 | 83.55 | 11.77 |
| FTO/PCBM/$MAPbI_3$/PEDOT: PSS/Ag | 1.11 | 13.66 | 82.71 | 12.58 |
| FTO/$SnO_2$/$MAPbI_3$/$TiO_2$:N/Ag | 1.08 | 11.33 | 82.00 | 10.11 |
| FTO/$SnO_2$/$MAPbI_3$/Spiro-OMeTAD/Ag | 1.11 | 11.28 | 81.62 | 10.23 |
| FTO/$SnO_2$/$MAPbI_3$/PEDOT: PSS/Ag | 1.10 | 11.31 | 86.84 | 10.89 |
| FTO/$SnO_2$/$MAPbI_3$/PTAA/Ag | 1.09 | 13.08 | 82.94 | 11.84 |

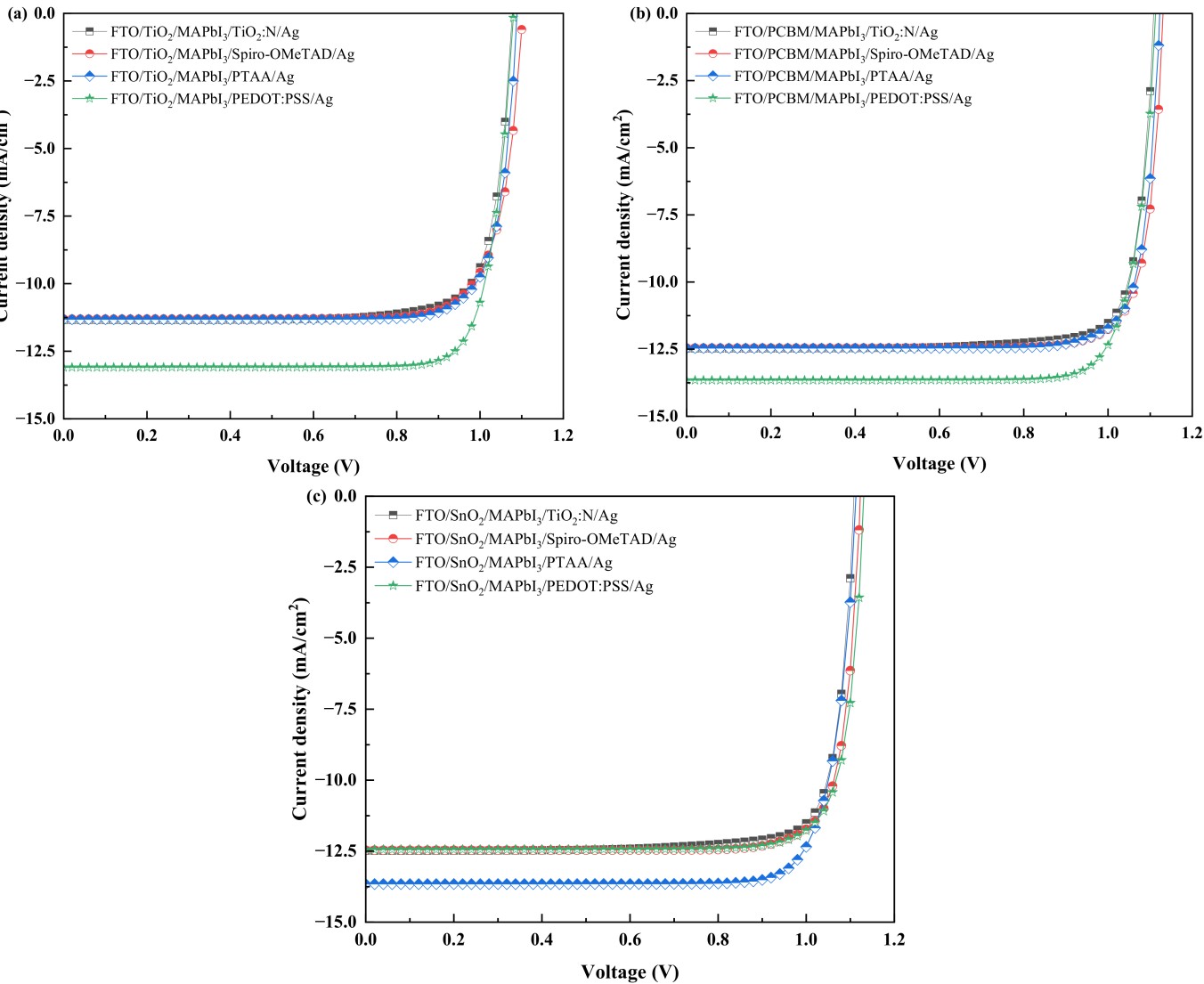

**Figure 3.** J-V characteristics of the PSC structures with four HTLs and the ETL as: (**a**) $TiO_2$. (**b**) PCBM. (**c**) $SnO_2$.

The results infer the significance of N-doped $TiO_2$ as an HTL with reasonable PCE attained compared to the other HTL structures. However, from Table 2, it is observed for all three cases of ETLs, $TiO_2$:N as an HTL delivered a lesser PCE (%) when compared to Spiro-OMeTAD, PEDOT: PSS and PTAA. Interestingly though, $TiO_2$:N as HTL depicted a PCE close to Spiro-OMeTAD and PTAA with a minor difference between 0.08% and 0.23% in the case of $TiO_2$ and PCBM as the ETL layer, while a comparatively higher drop in PCE of 1.83% and 1.04% was observed when equated to PEDOT: PSS as the HTL layer. In the case of $SnO_2$ as the ETL layer, simulation results depict a minor drop in PCE of 0.12% and 0.78% when employed with Spiro-OMeTAD and PEDOT: PSS as the HTL layers. However, PTAA as an HTL resulted in a PCE of 1.73% more than $TiO_2$:N, respectively. These results quantify the role of lesser effective density of states in conduction and valence bands and also the lower shallow acceptor density of $TiO_2$:N.

### 2.1. Influence of Absorber Layer Thickness on the Electrical Performance

From the previous section, it is prevalent that a $TiO_2$:N-based HTL is viable for producing efficiencies on par with polymeric HTLs. In this section, further numerical simulations are carried out to investigate the performance output for the proposed $TiO_2$:N-based HTL recipes when the absorber layer thickness varies beyond 100 nm. The PCE is

one of the vital factors expected to invariably respond to the degree of change in absorber layer thickness. Hence, this attempt would project the optimum and maximum thickness of the absorber layer to yield better output. The primary electrical parameters from Section 3.3 are considered while the absorber layer thickness is varied proportionally from 100 nm until the saturation level and maximum allowable thickness is observed. Simulations were carried out to analyze the electrical parameters when the HTL layer thickness was varied from 50 nm to 500 nm. The power conversion efficiency (PCE, %) and open circuit voltage ($V_{oc}$, V) were observed to have no change in their output, while minor differences were observed in the short circuit current ($J_{sc}$, mA/cm$^2$) and fill factor (FF, %). Hence, the least HTL thickness of 50 nm is considered for this simulation.

Conceptually, the increase in the absorber layer thickness influences the performance parameters of a PSC. With the increase in thickness, $J_{sc}$ tends to increase, since it is attributed to more electron-hole pairs in the absorber layer. In contrast, the $V_{oc}$ decreases due to the increment in the dark saturation current that increases the recombination of charge carriers. Seemingly, the FF holds an inversely proportional relationship with an increase in thickness due to the increase in series resistance and internal power dissipation. Lastly, the PCE increases with thickness but decreases beyond a saturation level (maximum diffusion length). Beyond the saturation point, the fill factor is reported to drop due sheet resistance of the active layer, which is a material perspective phenomenon [48,49].

The simulation results show the performance profile for all the three TiO$_2$:N-based HTL structures (FTO/TiO$_2$/MAPbI$_3$/TiO$_2$:N/Ag, FTO/PCBM/MAPbI$_3$/TiO$_2$:N/Ag and FTO/SnO$_2$/MAPbI$_3$/TiO$_2$:N/Ag) have depicted divergent patterns. Photovoltaic parameters report a change when the absorber layer thickness varies. The attained performance parameters for these recipes are tabulated in Table 3, with the patterns collectively presented in Figure 4. The results depict valid conceptual behavior in the aspect of electrical performance. For these recipes, 600 nm of absorber thickness was observed as the saturation point as indicated in Figure 5, beyond which the PCE and $V_{oc}$ were observed to drop gradually, mainly due to thermal recombination and sheet resistance of the absorber layer. For every interval of increase in absorber layer thickness, the recipe with PCBM as the ETL has been reported to produce higher outputs when compared to TiO$_2$ and SnO$_2$ based ETL recipes. In addition, numerical simulations were carried out to analyze these structures' performance when the HTL and ETL thickness was increased. Remarkably, the performance parameters remain unchanged with the increase in HTL thickness, but the increase in ETL thickness was observed to reduce the PCE by a fraction of 0.02% to 0.04%. Hence, the lower thickness of 50 nm was maintained for both the HTL and ETL throughout the simulation process.

**Table 3.** Performance parameters for both recipes with varied absorber thickness.

| PSC Structure | Thickness (nm) | $V_{oc}$ (V) | $J_{sc}$ (mA/cm$^2$) | FF (%) | PCE (%) |
|---|---|---|---|---|---|
| FTO/TiO$_2$/MAPbI$_3$/TiO$_2$:N/Ag | 100 | 1.07 | 11.35 | 81.09 | 9.92 |
| | 200 | 1.05 | 17.20 | 78.48 | 14.30 |
| | 300 | 1.04 | 20.48 | 77.26 | 16.54 |
| | 400 | 1.03 | 22.40 | 76.39 | 17.69 |
| | 500 | 1.02 | 23.58 | 75.50 | 18.24 |
| | 600 | 1.01 | 24.33 | 74.57 | 18.43 |
| | 700 | 1.00 | 24.82 | 73.54 | 18.41 |

**Table 3.** *Cont.*

| PSC Structure | Thickness (nm) | $V_{oc}$ (V) | $J_{sc}$ (mA/cm$^2$) | FF (%) | PCE (%) |
|---|---|---|---|---|---|
| FTO/PCBM/MAPbI$_3$/TiO$_2$:N/Ag | 100 | 1.10 | 12.48 | 83.32 | 11.54 |
| | 200 | 1.07 | 17.66 | 79.58 | 15.12 |
| | 300 | 1.05 | 20.64 | 77.64 | 16.93 |
| | 400 | 1.04 | 22.42 | 76.59 | 17.90 |
| | 500 | 1.03 | 23.53 | 75.74 | 18.37 |
| | 600 | 1.02 | 24.25 | 74.81 | 18.53 |
| | 700 | 1.01 | 24.72 | 73.83 | 18.49 |
| FTO/SnO$_2$/MAPbI$_3$/TiO$_2$:N/Ag | 100 | 1.08 | 11.33 | 82.00 | 10.11 |
| | 200 | 1.06 | 17.19 | 78.95 | 14.46 |
| | 300 | 1.04 | 20.47 | 77.42 | 16.64 |
| | 400 | 1.03 | 22.40 | 76.49 | 17.77 |
| | 500 | 1.02 | 23.58 | 75.61 | 18.31 |
| | 600 | 1.01 | 24.32 | 74.59 | 18.53 |
| | 700 | 1.01 | 24.81 | 73.68 | 18.48 |

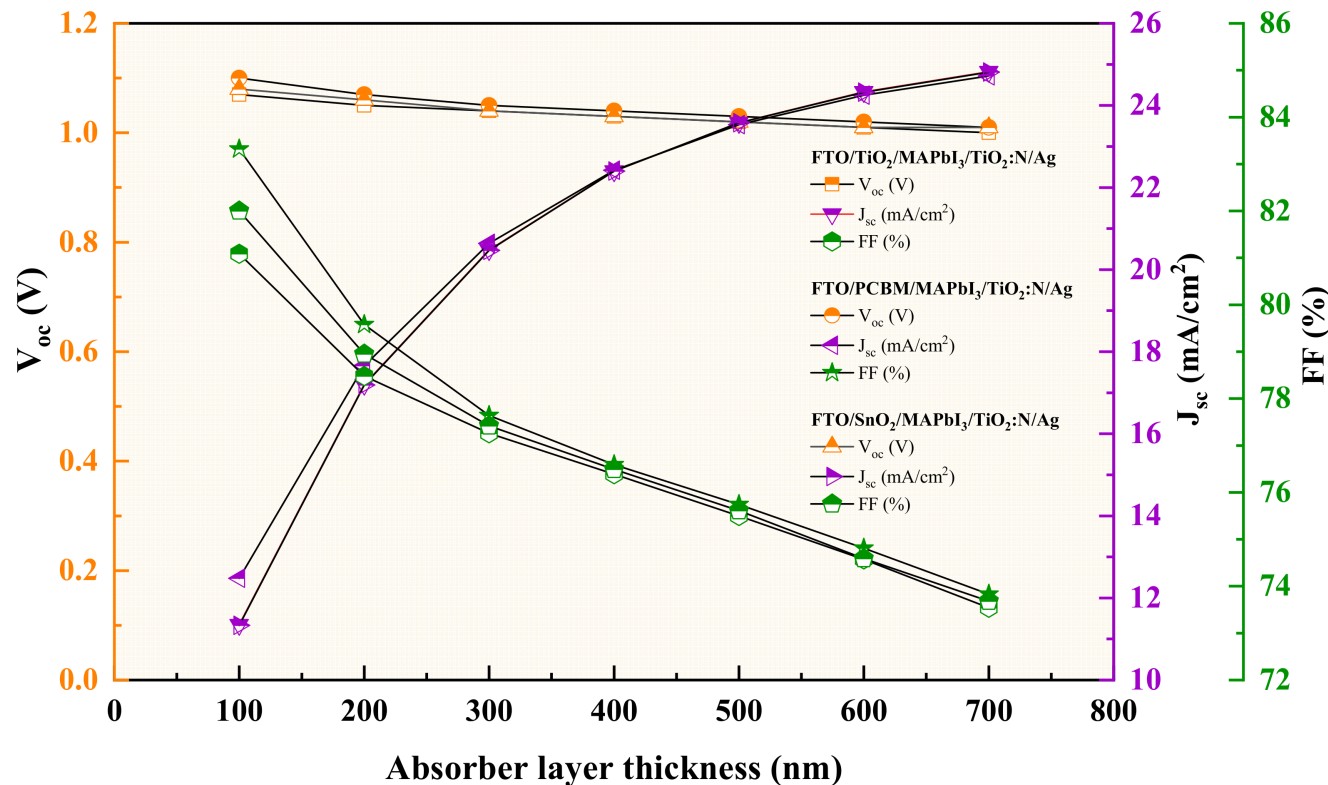

**Figure 4.** Patterns of the performance parameters for different absorber layer thickness.

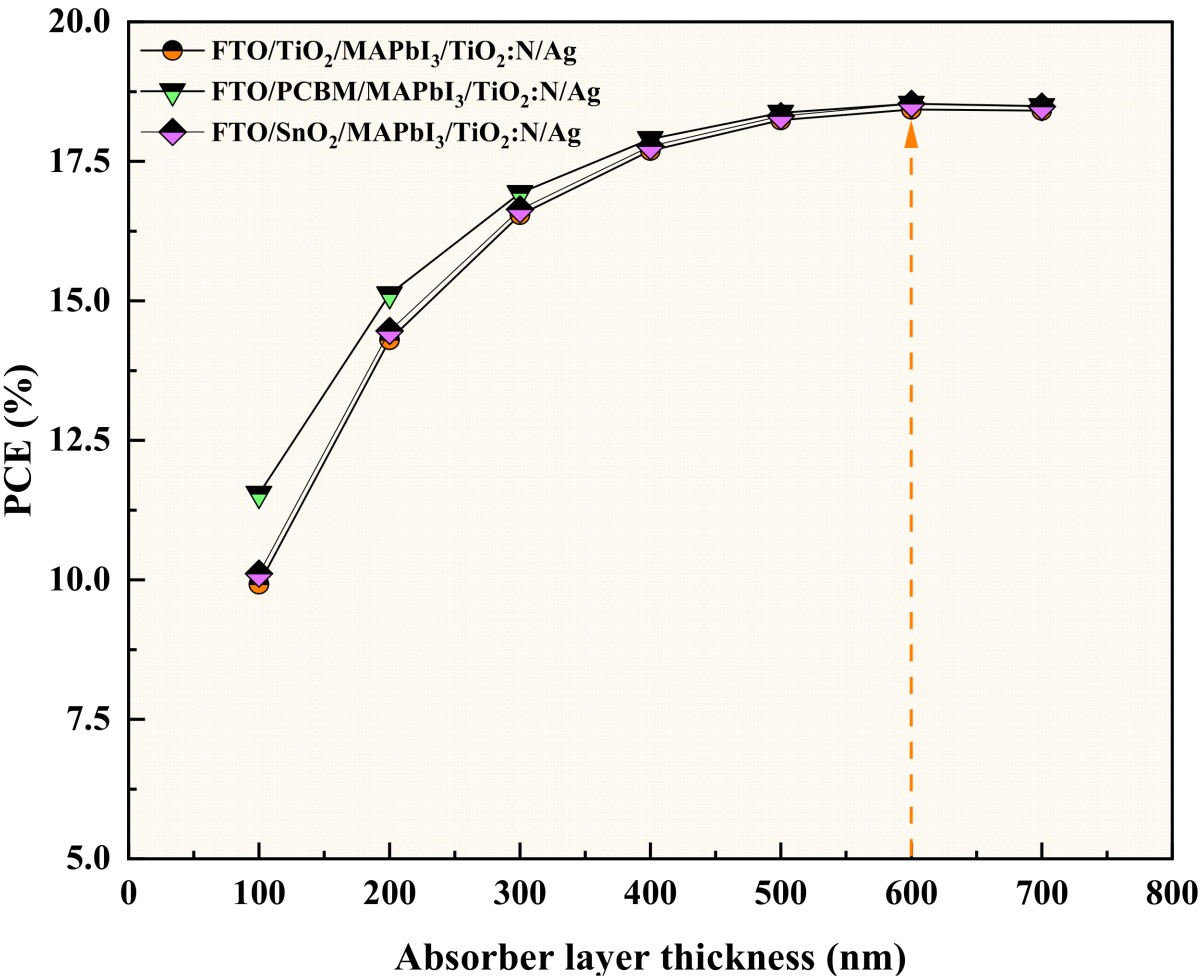

**Figure 5.** Efficiency curves for a gradient in absorber layer thickness indicating the threshold limit of 600 nm.

### 2.2. Influence of Absorber Layer Defect Density on the Electrical Performance

The quality and structure of the PSC absorber layer plays a significant role in delivering efficient power outputs. The performance parameters are influenced by the defect density ($N_t$) of the absorber layer that correlates to the film quality deterioration. This phenomenon causes a density trap and rise in the recombination of charge carriers, which reflects on the cell output [50]. Madan et al. [51] described that the defect densities on the perovskite/ETL and the perovskite/HTL directly influence the performance of the PSC. The effect is more intense when light is illuminated from the ETL side due to the high rate of photons being absorbed near the perovskite/ETL interface. This research investigates and reports the impact of defect density from the perovskite/ETL side as the illumination is projected from the ETL side. Hence, this simulation studies the performance of both the proposed perovskite recipes with a gradient in defect density ($N_t$) from $1 \times 10^{13}$ to $1 \times 10^{17}$ cm$^{-3}$ for the lower absorber thickness of 100 nm.

The J-V characteristics for the three recipes are reported in Figure 6a–c. Furthermore, the results of this simulation notify a consequent drop in the performance output parameters as the absorber defect density was increased, as reported in Table 4. The simulation results for both recipes infer the impact of the $N_t$ on the PCE (%). With the increase in defect density by $1 \times 10^1$ cm$^{-3}$, the $J_{sc}$, FF and $V_{oc}$ were observed to reduce logarithmically, resulting in a drop in the PCE (%) in the three recipes. For every degree of rising in the "$N_t$" by $1 \times 10^1$ cm$^{-3}$, the average efficiency drop was observed to be 1.84% for the FTO/TiO$_2$/MAPbI$_3$/TiO$_2$:N/Ag structure, 2.29% for the FTO/PCBM/MAPbI$_3$/TiO$_2$:N/Ag

structure and 1.90% for the FTO/SnO$_2$/MAPbI$_3$/TiO$_2$:N/Ag structure. This pattern infers the disadvantage of using PCBM as the ETL compared to TiO$_2$:N and SnO$_2$. However, it is observed that the increase in defect density has a lesser influence on the J$_{sc}$, but the V$_{oc}$ was observed to respond in proportional change. Interestingly, the J$_{sc}$ remained constant for the N$_t$ of 1 × 10$^{15}$ cm$^{-3}$ for both recipes. Beyond 1 × 10$^{15}$ cm$^{-3}$ of defect density, these recipes have reported a gradual drop in the electrical parameters reflected in the PCE (%). Henceforth, these simulations conclude the threshold for the absorber layer defect density is 1 × 10$^{15}$ cm$^{-3}$ for these respective recipes. The drift in the patterns plotted in Figure 6a–c; notifies that the converging trend is observed for a defect density of 1 × 10$^{15}$ cm$^{-3}$ and further.

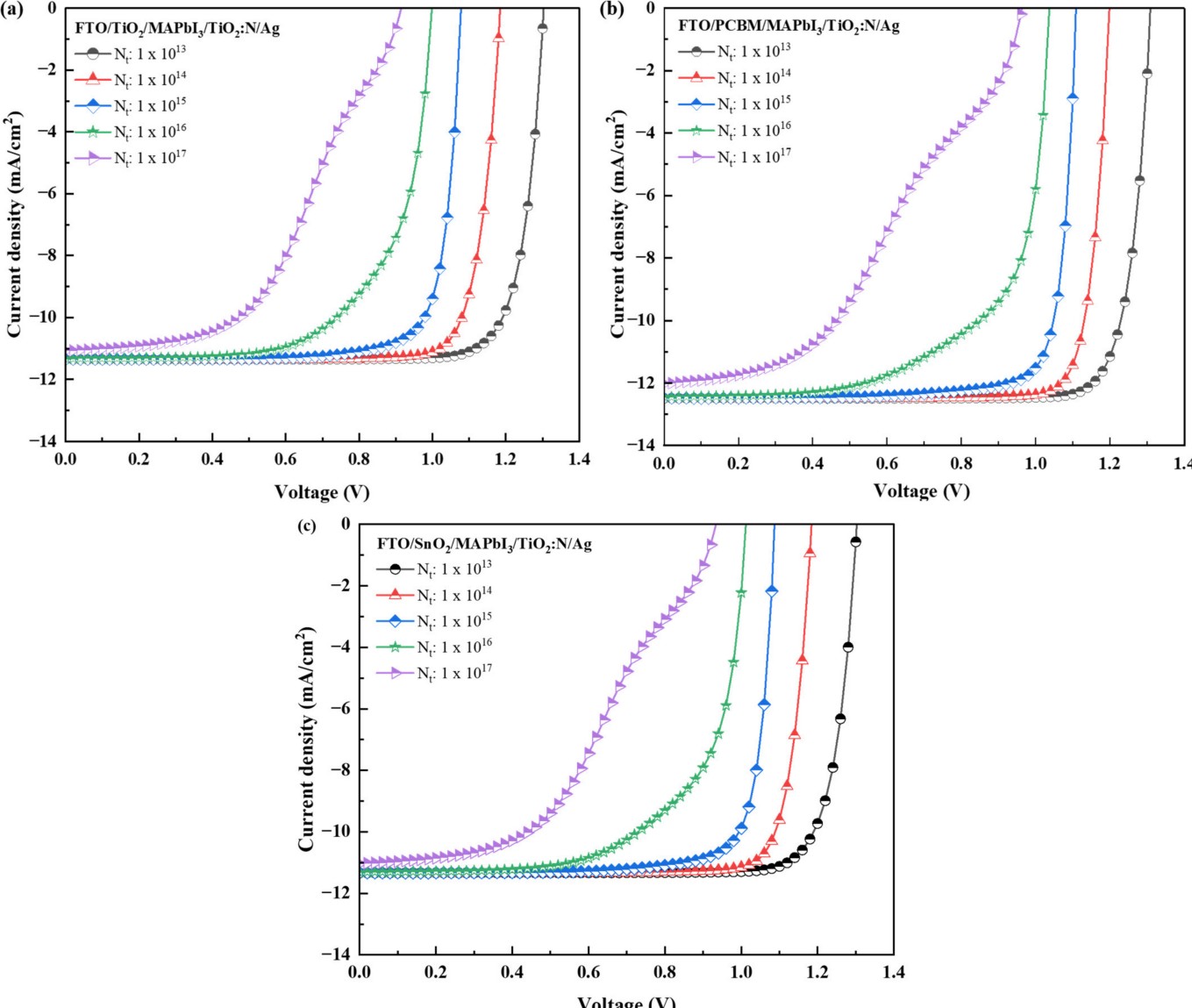

**Figure 6.** J-V characteristics of the PSC structures with TiO$_2$:N as HTL and TiO$_2$ as ETL (**a**); PCBM as ETL (**b**); and SnO$_2$ as ETL (**c**) layers.

**Table 4.** Performance parameters for proposed recipes with a gradient in defect density.

| Recipe | Absorber Layer Defect Density ($N_t$, cm$^{-3}$) | $V_{oc}$ (V) | $J_{sc}$ (mA/cm$^{-2}$) | FF (%) | PCE (%) |
|---|---|---|---|---|---|
| FTO/TiO$_2$/MAPbI$_3$/TiO$_2$:N/Ag | $1 \times 10^{13}$ | 1.30 | 11.35 | 83.42 | 12.34 |
| | $1 \times 10^{14}$ | 1.18 | 11.35 | 83.74 | 11.26 |
| | $1 \times 10^{15}$ | 1.07 | 11.35 | 81.09 | 9.92 |
| | $1 \times 10^{16}$ | 0.99 | 11.32 | 65.63 | 7.42 |
| | $1 \times 10^{17}$ | 0.91 | 11.07 | 49.18 | 4.98 |
| FTO/PCBM/MAPbI$_3$/TiO$_2$:N/Ag | $1 \times 10^{13}$ | 1.30 | 12.48 | 84.79 | 13.86 |
| | $1 \times 10^{14}$ | 1.19 | 12.48 | 85.87 | 12.85 |
| | $1 \times 10^{15}$ | 1.10 | 12.48 | 83.32 | 11.54 |
| | $1 \times 10^{16}$ | 1.03 | 12.43 | 66.05 | 8.52 |
| | $1 \times 10^{17}$ | 0.96 | 12.02 | 40.62 | 4.70 |
| FTO/SnO$_2$/MAPbI$_3$/TiO$_2$:N/Ag | $1 \times 10^{13}$ | 1.30 | 11.33 | 83.69 | 12.36 |
| | $1 \times 10^{14}$ | 1.18 | 11.33 | 84.80 | 11.39 |
| | $1 \times 10^{15}$ | 1.08 | 11.33 | 82.00 | 10.11 |
| | $1 \times 10^{16}$ | 1.02 | 11.30 | 65.08 | 7.45 |
| | $1 \times 10^{17}$ | 0.93 | 11.03 | 45.99 | 4.74 |

Additionally, further numerical simulations were carried out to study the influence caused on the efficiency of the defect density of HTL and ETL layers. The $N_t$ was varied from $1 \times 10^3$ to $1 \times 10^7$ cm$^{-3}$ for the transport layers. The simulation results have reported no change in the performance of either PSC structure. This denotes the influence of defect density susceptible to the perovskite absorber layer only.

## 3. Methodology

### 3.1. SCAPS Software Simulation

The solar cells capacitance simulator (SCAPS 3.10) one-dimensional software is a modern computational tool developed to simulate solar cell physics numerically. SCAPS provides a theoretical understanding of solar cell behavior that helps compare results with experimental analysis [52]. The SCAPS built-in program is designed to numerically solve semiconductor equations in 1D steady-state conditions. Global researchers recognized SCAPS as a suitable analytical tool to determine I-V characteristics, fill factor, band diagrams, quantum efficiencies, spectral responses, short-circuit current and open-circuit voltage, PCE and recombination profile within the charge transport layers [53–55]. The performance of the fabricated solar cell is based on the semiconductor Equations (1)–(6);

$$Electron\ continuity\ eqn., \quad \frac{dn_p}{dt} = G_n - \frac{n_p - n_{po}}{\tau_n} + n_p\, \mu_n \frac{d\xi}{dx} + \mu_n \xi \frac{dn_p}{dx} + D_n \frac{d^2 n_p}{dx^2} \quad (1)$$

$$Hole\ continuity\ eqn., \quad \frac{dp_n}{dt} = G_p - \frac{p_n - p_{no}}{\tau_p} - p_n\, \mu_p \frac{d\xi}{dx} - \mu_p \xi \frac{dp_n}{dx} + D_p \frac{d^2 p_n}{dx^2} \quad (2)$$

$$Poissons\ eqn., \quad \frac{d}{dx}\left(-\varepsilon(x)\frac{d\psi}{dx}\right) = q\left[p(x) - n(x) + N_d^+(x) - N_a^-(x) + p_t(x) - n_t(x)\right] \quad (3)$$

On a more fundamental basis, an alternative way to express the equations was presented by [56]. Equation (3) is also described as;

$$\frac{d^2}{dx^2}\psi(x) = \frac{q}{\varepsilon}\left[p(x) - n(x) + N_D - N_A + p_h - p_e\right] \quad (4)$$

$$\frac{dJ_n}{dx} = G - R \quad (5)$$

$$\frac{dJ_p}{dx} = G - R \qquad (6)$$

### 3.2. TiO₂:N as a p-Type HTL

As an alternative to expensive HTLs, this research examines the applicability and feasibility of implementing $TiO_2$:N as an HTL in a PSC recipe. Several studies revealed that N-doped $TiO_2$ exhibits stable p-type conductivity. Towards analyzing the cation vacancies in $TiO_2$, Lee et al. [57] reported that n-type $TiO_2$ and p-type $TiO_2$ exhibit similar morphology, surface area and crystal structure. Comparatively, p-type $TiO_2$ has better stability and performance rate. Vasu et al. [58] employed the atomic layer deposition technique to develop a p-type epitaxial N-doped $TiO_2$ thin film. The results depict a reduced optical bandgap and better hole concentration and mobility. Vasilopoulou et al. [59] stated that p-type nitrogen doping enhances the photocatalytic efficiency of $TiO_2$ in the visible spectrum, while Anitha et al. [60] reported that the charge transportation could be eased in $TiO_2$ due to additional bands, which can be achieved through cationic doping. Researchers synthesized the p-type $TiO_2$:N film with different nitrogen doping concentrations. As the N-doping increases, results depicted a change in light transmittance (%), optical bandgap ($E_g$), Hall coefficient ($cm^3/C$), carrier density ($cm^{-3}$), conductivity ($\Omega^{-1}.cm^{-1}$) and mobility ($cm^2.v^{-1}.s^{-1}$). In addition, a further increase in nitrogen concentration would lead to a short-circuit in the cell, making it inappropriate as a photocathode. Literature studies inferred that $TiO_2$ has a tunable bandgap nature that depends on the nitrogen concentration [59,60].

### 3.3. The Proposed Perovskite Solar Cell Structure

This research simulates a PSC recipe developed with methylammonium lead iodide ($MAPbI_3$) as the absorber layer. Various charge transport layers are considered to investigate the performance, and the optimum high responsive match in electrical performance is reported. Certain characteristics of the MA-based absorber layer include the ability of lower thickness, lower bandgap of 1.5 eV, lower defect states, better thermal stability and good optical properties compared to other absorber layers. Hence, MA-based PSCs are considered one of the most promising light-absorbing perovskite materials. Likewise, to validate the electrical behavior of $TiO_2$:N as an HTL, a comparative analysis is carried out in this simulation by adopting two different polymeric HTLs (Spiro-OMeTAD, PEDOT: PSS and PTAA) and ETLs ($TiO_2$, PCBM and $SnO_2$) which tend to possess efficient light transmission properties. These charge transport layers' electrical parameters are optimized according to the parameters adopted from earlier various experimental and simulation studies that reported their significance. SCAPS software simulation for the proposed structures is performed using distant parameters, reported in Table 5. In view of attaining thin-film light transmission solar cells, the thickness of the charge transport layers is considered 50 nm, as the lowest value, which is an attainable thickness from various coating techniques [61]. The planned n−i−p PSC configuration structure is illustrated in Figure 7a. In contrast, the SCAPS simulation design of the proposed structure is pictured in Figure 8. Adopting $TiO_2$ as HTL is a unique technique for developing a low-cost PSC with higher optical properties. Literature reports that the bandgap and light transmission (%) is significantly reduced as the doping concentration increases.

**Table 5.** The proposed electrical parameter considered for software simulation of PSC structures [62–71].

| Parameters | Substrate | | ETL | | Perovskite Absorber Layer | Novel HTL | Polymeric HTLs | | |
|---|---|---|---|---|---|---|---|---|---|
| | FTO | TiO$_2$ | PCBM | SnO$_2$ | MAPbI$_3$ | TiO$_2$:N | Spiro-OMeTAD | PEDOT: PSS | PTAA |
| Thickness '$t$' (nm) | 200 | 50 | 50 | 50 | 100 * | 50 | 50 | 50 | 50 |
| Bandgap '$E_g$' (eV) | 3.5 | 3.2 | 2 | 3.4 | 1.55 | 3 | 2.88 | 1.8 | 2.96 |
| Electron affinity '$\chi$' (eV) | 4 | 4 | 3.9 | 4 | 3.9 | 2.2 | 2.05 | 3.4 | 2.3 |
| Dielectric Permittivity '$\varepsilon_r$' | 9 | 9 | 4 | 9 | 6.5 | 3 | 3 | 3 | 9 |
| CB EDOS '$N_c$' (cm$^{-3}$) | $2.2 \times 10^{18}$ | $2.2 \times 10^{18}$ | $1 \times 10^{21}$ | $2.2 \times 10^{18}$ | $2.2 \times 10^{19}$ | $1.3 \times 10^{14}$ | $2.2 \times 10^{18}$ | $2.2 \times 10^{18}$ | $1 \times 10^{21}$ |
| VB EDOS '$N_v$' (cm$^{-3}$) | $2.2 \times 10^{18}$ | $1.8 \times 10^{19}$ | $2 \times 10^{20}$ | $1.8 \times 10^{19}$ | $1.3 \times 10^{19}$ | $1.3 \times 10^{15}$ | $1.8 \times 10^{19}$ | $1.8 \times 10^{19}$ | $1 \times 10^{21}$ |
| e$^-$ thermal velocity (cm.s$^{-1}$) | $1 \times 10^7$ | $1 \times 10^7$ | $1 \times 10^7$ | $1 \times 10^7$ | $1 \times 10^7$ | $1 \times 10^7$ | $1 \times 10^7$ | $1 \times 10^7$ | $1 \times 10^7$ |
| h$^+$ thermal velocity (cm.s$^{-1}$) | $1 \times 10^7$ | $1 \times 10^7$ | $1 \times 10^7$ | $1 \times 10^7$ | $1 \times 10^7$ | $1 \times 10^7$ | $1 \times 10^7$ | $1 \times 10^7$ | $1 \times 10^7$ |
| Electron mobility '$\mu_n$' (cm$^2$/V.s) | 20 | 20 | $1 \times 10^{-2}$ | 20 | 2.7 | 2 | $2 \times 10^{-4}$ | 100 | 1 |
| Hole mobility '$\mu_h$' (cm$^2$/V.s) | 10 | 10 | $1 \times 10^{-2}$ | 10 | 1.8 | 2 | $2 \times 10^{-4}$ | 4 | 40 |
| Shallow donor density '$N_D$' (cm$^{-3}$) | $2 \times 10^{19}$ | $1 \times 10^{16}$ | $1 \times 10^{20}$ | $1 \times 10^{17}$ | $1.3 \times 10^{16}$ | 0 | 0 | 0 | 0 |
| Shallow Acceptor density '$N_A$' (cm$^{-3}$) | 0 | 0 | 0 | 0 | $1.3 \times 10^{16}$ | $1.3 \times 10^{14}$ | $2 \times 10^{19}$ | $2 \times 10^{19}$ | $2 \times 10^{19}$ |
| Defect density '$N_t$' (cm$^{-3}$) | $1 \times 10^{15}$ | $1 \times 10^{15}$ | $1 \times 10^{14}$ | $1 \times 10^{15}$ | $1 \times 10^{15}$ * | $1 \times 10^{15}$ | $1 \times 10^{15}$ | $1 \times 10^{14}$ | $1 \times 10^{15}$ |

* Varied parameter.

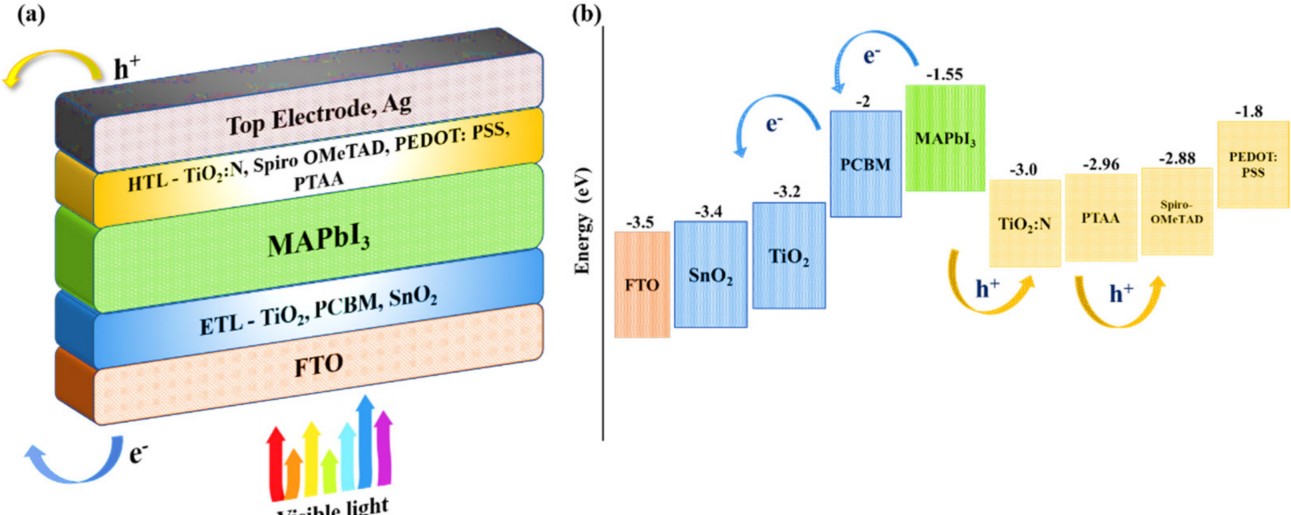

**Figure 7.** (**a**) Proposed n−i−p structure of the PSC with different layers. (**b**) Energy band diagram of the PSC structure (combined) with bandgap denotation for absorber and charge transport layers [62–71].

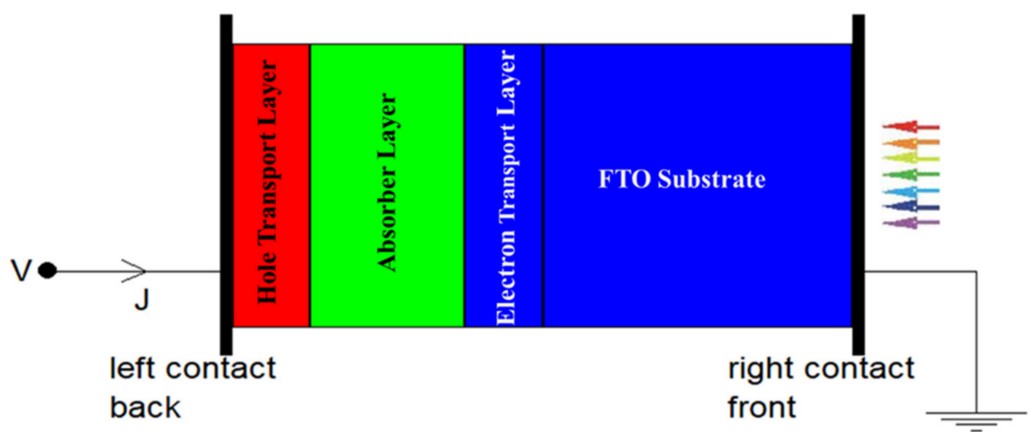

**Figure 8.** SCAPS simulation design of the PSC structure.

The bandgap of $TiO_2$:N was optimized by Vasu et al. [58] to develop a p-type epitaxial N-doped $TiO_2$ thin film. The initial bandgap of bulk anatase n-type $TiO_2$ was observed to be 3.23 eV; upon doping with 4.0% concentration (from XPS analysis) of nitrogen, the bandgap was deemed to reduce to 3.07 eV in the $TiO_2$:N film which also reported p-type behavior due to the decrease in the fermi-energy levels. Additionally, from preliminary simulations performed in this research, it was observed that an increase in nitrogen concentration would induce a short-circuit in the cell when the HTL bandgap is less than 2.5 eV. Hence, nitrogen doping levels must be controlled to attain a bandgap of more than 2.5 eV for an ideal HTL medium with optimum light transmission properties. Therefore, considering previous experimental studies, this study investigates the behavior of the PSC structure with a proposed HTL bandgap of 3.0 eV. The energy band diagram for the proposed corresponding layers in the PSC structure is illustrated in Figure 7b. The energy levels in Figure 7b were optimized and reported according to previous experimental data.

## 4. Conclusions

Among the strategies identified from the reported literature, this study deals with simulating a PSC structure with an absorber layer having a lower bandgap and layer thickness as well as a novel inorganic charge transport medium in the form of N-doped $TiO_2$. This work proposes the optimal electrical parameters for different MA-based perovskite recipes that can attain rational outputs. SCAPS-1D software simulation has been performed in this study with three specific PSC structures: $FTO/TiO_2/MAPbI_3/TiO_2$:N/Ag, $FTO/SnO_2/MAPbI_3/TiO_2$:N/Ag and $FTO/PCBM/MAPbI_3/TiO_2$:N/Ag, respectively. The present results were validated by simulating the PSC structure with different polymeric HTL and ETL layers available in the literature. The simulation results reveal that the proposed PSC structures with $TiO_2$:N as an HTL have estimated a PCE of 9.92%,10.11% and 11.54%, which are on par with other PSC structures employing polymeric HTLs. Simulations reveal that $TiO_2$:N as an HTL can deliver a PCE on par with Spiro-OMeTAD and PTAA when $TiO_2$ and PCBM are used as the ETL layers, while it closely matches the PCE of PEDOT: PSS when $SnO_2$ is employed as the ETL medium. Furthermore, the absorber layer thickness for the proposed recipes was varied from 100 nm to 700 nm in steps of 100 nm, and it was observed that 600 nm was the threshold layer thickness that avoids recombination losses. In addition, the influence of absorber layer defect density on the performance of PSC was analyzed to identify the saturation limit through the drift pattern. Defect density beyond $1 \times 10^{15}$ $cm^{-3}$ observed a drop in $J_{sc}$ and, thereby, a reduction in the fill factor. Lastly, the electrical parameters of the proposed mixed metal halide PSC structures with $TiO_2$:N as an HTL could be used to envisage a semi-transparent perovskite solar cell with the lowest layer thickness, reasonable efficiencies and comparatively lower costs.

**Author Contributions:** Conceptualization, N.R.P. and Y.R.S.; methodology, N.R.P. and Y.R.S.; validation, N.R.P. and Y.R.S.; investigation, Y.R.S.; writing—original draft preparation, N.R.P.; review and editing, N.R.P. and Y.R.S.; supervision, Y.R.S. All authors have read and agreed to the published version of the manuscript.

**Funding:** The funding was supported by Vellore Institute of Technology University.

**Data Availability Statement:** Not applicable.

**Acknowledgments:** The authors thank Marc Burgelman, ELSI, University of Gent, Belgium, for providing the SCAPS 3.10 version simulation software used in this numerical study. The authors thank Vellore Institute of Technology for supporting the open-access publication fee.

**Conflicts of Interest:** The authors declare no conflict of interest.

**Sample Availability:** Samples of the compounds are available from the authors.

## Nomenclature

| | | | |
|---|---|---|---|
| $\Psi$: | electrostatic potential | $p_h$: | hole distribution |
| $q$: | electric charge | $p_e$: | electron distribution |
| $\varepsilon$: | dielectric constant | $J_n$: | current density for electron |
| $p$: | hole concentration | $J_p$: | current density for the hole |
| $n$: | electron concentration | $G$: | Generation rates for carriers |
| $N_D$: | doping concentration for the donor | $R$: | recombination rates for carriers |
| $N_A$: | doping concentration for acceptor | | |

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
