# Peer review of "Numerical Simulation of Nitrogen-Doped Titanium Dioxide as an Inorganic Hole Transport Layer in Mixed Halide Perovskite Structures Using SCAPS 1-D"

_inorganics, doi:10.3390/inorganics11010003_

Round 1
Reviewer 1 Report
The authors used SCAPS simulation software to explore the electrical performance of perovskite solar cells (PCSs) with different transporting materials, which aimed to envisage the conceptuality in devising semi-transparent PSCs. For the HTL, they compared N doped TiO2, PEDOT:PSS and Spiro-OMeTAD. For the ETL, they compared TiO2 and PCBM. In addition, they compared film thickness and defect density of perovskite layer. In all, lots of contents have been discussed in this manuscript. However, scientific findings are quite few, with poor simulation design. As a result, I cannot recommend this manuscript published on Inorganics. Detailed comments are as follows:
1. N doped TiO2 was compared with PEDOT:PSS and Spiro-OMeTAD. However, the PEDOT:PSS is used less frequently than NiO and PTAA at present, as PEDOT:PSS is easy to absorb water and thus accelerates degradation of halide perovskites. Spiro-OMeTAD also has similar problem. So, NiO and PTAA are recommended to be compared instead of PEDOT:PSS and Spiro-OMeTAD.
2. The authors used FTO/PCBM/MAPbI3/TiO2:N/Ag structure for simulation. However, PCBM is rarely used between FTO/MAPbI3, as high quality MAPbI3 could be hardly deposited on PCBM. Instead, SnO2 should be included instead of PCBM.
3. In addition to the improper structures mentioned above, I recommend to compare the structures, such as FTO/TiO2:N/MAPbI3/PCBM/Ag, FTO/NiO/MAPbI3/PCBM/Ag and FTO/PTAA/MAPbI3/PCBM/Ag. In this way, the author could tell whether TiO2:N is better than PTAA or NiO.
4. Nowadays, FAPbI3 and mixed-cation halide perovskites have been shown much higher power conversion efficiency than MAPbI3. Considering a simulation should provide guidance to experiments, FAPbI3 and mixed-cation halide perovskites should be simulated instead of MAPbI3.
5. In experiments, film thickness and trap density of halide perovskites have been optimized, which yields over 25% power conversion efficiency of PSCs. The author is strongly suggested to simulate TiO2:N in a high efficiency PSC model, but not PSCs with PCE ≤ 20%.
Reviewer 2 Report
In the paper, “Numerical simulation of Nitrogen-doped titanium dioxide as an inorganic hole transport layer in mixed halide perovskite structures using SCAPS 1-D”, the authors presented a study on the electrical performance of different Methylammonium (MA) based perovskite structuresthat can envisage the conceptuality in devising ST PSCs. I commend the authors for their work and I recommend the manuscript published with the following revisions. Please check: 1.How the PCE efficiency is calculated? 2.I suggest that the authors should point that the experimental or theoretical results in Table 1. 3.Some relative references are suggested, such as J. Phys. D: Appl. Phys. 2022, 55, 293002; Materials today 18.2 (2015): 65-72; ACS Energy Letters, 2017, 2(4): 802-806.4.Why does FTO/TiO2/MAPbI3/TiO2:N/Ag or FTO/PCBM/MAPbI3/TiO2:N/Ag achieve the maximum PCE efficiency at the thickness of 600 nm.
Reviewer 3 Report
Referee report of manuscript with ID 2026283
In this manuscript the authors present result of simulations of a perovskite solar cell (PSC) structure with an absorber layer characterized by a lower band gap and layer thickness in the form of N-doped TiO2.The simulations were performed employing the SCAPS-1D software. The applied methodology was validated with data available in the literature.
The manuscript is well presented and the work is interesting.
From this numerical simulation of Nitrogen-doped titanium dioxide as an inorganic hole transport layer in mixed perovskite structures, the authors provide detailed information of power conversion efficiency and maximum allowable absorber thickness. These data that could be useful to develop thin-film light transmission perovskite cells with reasonable efficiencies and low layer thickness.
I would suggest the authors to include in section 2.1 a more detailed description concerning the semiconductor equations on which the SCAPS-1D software is based. This is useful for the readers who are not very familiar with this program.
Reviewer 4 Report
Dear Authors,
My comments can be found in the .docx file attached below.
Comment 1:
The authors claim in the abstract that: “The influence of absorber thickness, defect density and hole-electron mobilities are analysed with optimal parameters.” While the influence of absorber thickness and defect density are analysed in chapters 3.1 and 3.2, respectively, there are no data about the hole-electron mobilities.
Comment 2:
In paragraph 2.1, lines 179-180: what is “current-voltage density”? SCAPS can calculate I-V characteristics, but can’t calculate “current-voltage density”.
Comment 3:
Using the parameters in Table 2., the authors have simulated the performance of PSC and the results are presented in Table 3. However, I did the same simulations (for only three structures) and get some different results presented in Table X below. I used SCAPS (ver. 3.3.09) with no resistance, as it is mentioned in the text, and standard test conditions (AM1.5G solar spectrum with the intensity of 1000 W/m2 and temperature of 300 K). Is there any extra parameter entered in the calculation that was not mentioned in the text? How the Ag electrode was characterised in the simulations?
Table X.
|
Device structure |
Voc (V) |
Jsc (mA/cm2) |
FF (%) |
PCE (%) |
|
FTO/TiO2/MAPbI3/TiO2:N/Ag |
0.83 |
11.31 |
55.73 |
5.21 |
|
FTO/TiO2/MAPbI3/Spiro-OMeTAD/Ag |
1.10 |
11.29 |
71.24 |
8.87 |
|
FTO/PCBM/MAPbI3/ TiO2:N /Ag |
0.86 |
12.43 |
59.99 |
6.44 |
Comment 4:
Fig.5 (a, b) have Jsc and Voc as physical parameters on their axis. These parameters should be J and V, and their names must be corrected in the text also (lines 262) and Fig. 5. (line 267). The same correction should be done for Fig.8., too.
Comment 5:
Since all the graphs have physical units inside the brackets, please correct it for the Voc in Fig.7.
Overall, I suggest rejecting this work in its current form because the simulations are not conducted correctly.
Round 2
Reviewer 1 Report
The authors responsed all the comments properly and added simulation results to improve the overall scientific importance of this manuscript. I have no further comment.
Author Response
The authors thank the reviewer for considering revisions made in the revised manuscript.
The authors hereby intimate that extensive English language check has been carried out in the current revised manuscript.
Reviewer 4 Report
Dear authors,
I have two new Comments:
1. Instead of "current density and voltage" (lines 181-182), I suggest changing it to "short circuit current and open circuit voltage". These are the important parameters that SCAPS can determine.
2. In the revised version, there are 79 references in total, but in the text, there are only 74 mentioned. Please correct it.
